# Molecular Events in the Melanogenesis Cascade as Novel Melanoma-Targeted Small Molecules: Principle and Development

**DOI:** 10.3390/cancers14225588

**Published:** 2022-11-14

**Authors:** Kazumasa Wakamatsu, Akira Ito, Yasuaki Tamura, Tokimasa Hida, Takafumi Kamiya, Toshihiko Torigoe, Hiroyuki Honda, Shosuke Ito, Kowichi Jimbow

**Affiliations:** 1Institute for Melanin Chemistry, Fujita Health University, Toyoake 470-1192, Aichi, Japan; 2Department of Chemical Systems Engineering, School of Engineering, Nagoya University, Nagoya 464-8603, Aichi, Japan; 3Department of Pathology, Sapporo Medical University School of Medicine, South 1 West 16, Chuo-ku, Sapporo 060-8556, Hokkaido, Japan; 4Department of Dermatology, Sapporo Medical University School of Medicine, Sapporo 060-8556, Hokkaido, Japan; 5Department of Biomolecular Engineering, School of Engineering, Nagoya University, Nagoya 464-8603, Aichi, Japan; 6Institute of Dermatology & Cutaneous Sciences, 1-27 Odori West 17, Chuo-ku, Sapporo 060-0042, Hokkaido, Japan

**Keywords:** antimelanoma targeted approach, melanogenesis metabolites, targeting with small molecules, magnetite nanomedicine, chemo-thermo-immunotherapy

## Abstract

**Simple Summary:**

Melanin biosynthesis can be a unique pathway to identify novel anti-melanoma targeted molecules. In this approach, we specifically focused on the substrate of tyrosinase, which is highly expressed in malignant melanoma. Among the various forms of melanogenesis substrate synthesized, *N*-propionyl cysteaminylphenol was exploited to develop a melanoma-targeted chemo-immunotherapy drug because of its selective incorporation into melanoma cells and production of highly reactive molecules, which not only result in apoptotic cell death but also the generation of heat shock proteins by reacting with tyrosinase. Moreover, the drug was attached to magnetite nanoparticles in order to enable the heating of melanoma cells when they are exposed to an alternating magnetic field, which causes non-apoptotic cell death and further heat-shock protein generation (thermo-immunotherapy). Here, we review our synthesis of melanogenesis-based anti-melanoma molecules and development of selective chemo-thermo-immuno-therapy by combining these molecules with the magnetite-nanoparticles. We compare this strategy to other melanogenesis-based chemotherapy and thermal medicine systems, and discuss targeted therapies with immune checkpoint inhibitors for unresectable/metastatic melanoma.

**Abstract:**

Malignant melanoma is one of the most malignant of all cancers. Melanoma occurs at the epidermo–dermal interface of the skin and mucosa, where small vessels and lymphatics are abundant. Consequently, from the onset of the disease, melanoma easily metastasizes to other organs throughout the body via lymphatic and blood circulation. At present, the most effective treatment method is surgical resection, and other attempted methods, such as chemotherapy, radiotherapy, immunotherapy, targeted therapy, and gene therapy, have not yet produced sufficient results. Since melanogenesis is a unique biochemical pathway that functions only in melanocytes and their neoplastic counterparts, melanoma cells, the development of drugs that target melanogenesis is a promising area of research. Melanin consists of small-molecule derivatives that are always synthesized by melanoma cells. Amelanosis reflects the macroscopic visibility of color changes (hypomelanosis). Under microscopy, melanin pigments and their precursors are present in amelanotic melanoma cells. Tumors can be easily targeted by small molecules that chemically mimic melanogenic substrates. In addition, small-molecule melanin metabolites are toxic to melanocytes and melanoma cells and can kill them. This review describes our development of chemo-thermo-immunotherapy based on the synthesis of melanogenesis-based small-molecule derivatives and conjugation to magnetite nanoparticles. We also introduce the other melanogenesis-related chemotherapy and thermal medicine approaches and discuss currently introduced targeted therapies with immune checkpoint inhibitors for unresectable/metastatic melanoma.

## 1. Introduction: Overall View of Melanogenesis Cascade to Develop Novel Antimelanoma Approaches by Exploiting Melanogenesis-Based Small Molecules

Malignant melanoma possesses a unique metabolic pathway, the synthesis of melanin pigments, which are formed by the conversion of tyrosine to dopaquinone in the presence of tyrosinase (EC 1.14.18.1). This process occurs in specific secretory granules: melanosomes [1]. Different synthetic stages of melanin pigments have been exploited to develop melanoma cell-specific chemotherapeutic agents [2]. Most previous attempts have utilized dopa and related catechol compounds, which cause general cytotoxicity through autoxidation [3].

Tyrosine analogues, which are tyrosinase substrates, are, however, excellent candidates for melanoma-specific targeting therapy [4]. Melanogenesis, a biochemical process unique to melanocytes; is highly expressed in malignant melanoma. A specific enzyme tyrosinase catalyzes the oxidative conversion of L-tyrosine to melanin pigments in melanocytes and malignant melanoma cells [5]. Harnessing melanogenesis to develop melanoma-specific antitumor agents has been a challenging goal [6]. Tyrosinase can oxidize a variety of natural and synthetic phenols to produce highly reactive and cytotoxic *o*-quinones [7,8]. Thus, we designed synthetic molecules for the selective treatment of unresectable/metastatic melanoma based on the melanocyte- and melanoma-cell-specific metabolic process melanogenesis (Table 1) [9,10,11,12,13,14,15,16,17,18,19,20,21,22,23,24,25,26,27,28,29,30].

**Table 1 cancers-14-05588-t001:** Phenolic thioethers synthesized and evaluated for antimelanoma and depigmenting effects.

Compound	Abbreviation	Structure	Synthesis	Tyrosinase Substrate ^a^	In Vitro Cytotoxicity ^b^	In Vivo Antimelanoma Effect ^c^	In Vivo Depigmentation ^d^
4-*S*-Cysteinylphenol	4SCP	1	[9]	Yes [16]	[9,21]	Yes/No [9,10,16]	Yes/No [16,28]
2-*S*-Cysteinylphenol	2SCP	2	[9]	No [16]	[9]		No [16]
4-*S*-Cysteinylcatechol	4SCC	3	[9]	Yes [16]	[9,21]	No [9,10]	No [16]
3-*S*-Cysteinylcatechol	3SCC	4	[9]		[9,21]	No [10]	
2-*S-*Cysteinylhydroquinone	2SCHQ	5	[10]	No [16]	[21]	No [10]	No [16,28]
4-*S*-Cysteaminylphenol	4SCAP	6	[10]	Yes [16,17]	[11,21,22]	Yes [10,16,23,24,25]	Yes [16,23,29]
2-*S*-Cysteaminylphenol	2SCAP	7	[10]	Yes [16]		No [10]	
4-*S*-Homocysteaminylphenol	4SHCAP	8	[11]	Yes [17]	[11]		Yes [29]
4-*S*-α-Methylcysteaminylphenol	4SMeCAP	9	[11]	Yes [17]	[11]		Yes [29]
*N*,*N*-Dimethylcysteaminylphenol	*N*,*N*-DiMeCAP	10	[11]	Yes [17]	[11]		Yes [29]
(*R*)- or (*S*)-4-*S*-α-Methylcysteaminylphenol	*R,S*-4SMeCAP	11	[12]	Yes [12]	[12]	Yes [26]	Yes [26]
(*R*)- or (*S*)-4-*S*-α-Ethylcysteaminylphenol	*R,S*-4SEtCAP	12	[12]	Yes [12]	[12]	Yes [26]	Yes [26]
4-*S*-Cysteaminylcatechol	4SCAC	13	[13]	Yes [13]	[13]	Yes [13]	
3-*S*-Cysteaminylcatechol	3SCAC	14	[13]	No [13]	[13]	No [13]	
2-*S*-Cysteiaminylhydroquinone	2SCAHQ	15	[13]	No [13]	[13]	No [13]	
*N*-Acetyl-4-*S*-cysteaminylphenol	NAcCAP	16	[14]	Yes [18]	[19]	Yes [24]	Yes [29,30]
*N*-Propionyl-4-*S*-cysteaminylphenol	NPrCAP	17	[15]	Yes [15,18,20]	[15,18,19]	Yes [15,27]	Yes [15,18]

^a^ Yes = Substrate; No = Not substrate. ^b^ Due to diversity of cell lines and methods used, comparison of the results is difficult. ^c^ Yes = Significant suppression of tumor volume, suppression of lung colony formation, or elongation of life span. No = No or small non-significant effects. ^d^ Yes = Partial to complete depigmentation of plucked hair after regrowth. No = No visible effects.



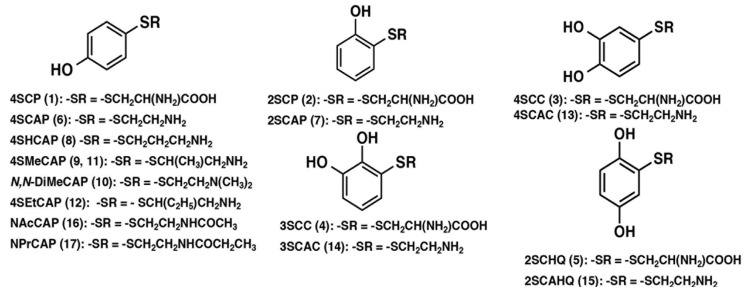



Utilizing the cytotoxic reaction in the melanogenesis cascade, we synthesized sulfur-containing phenolic amine derivatives of tyrosine to increase its affinity with the cell membrane and to destroy selected melanoma cells. Our previous in vivo studies showed the selective melanocyte-toxicity and anti-melanoma effects of new phenolic thioether compounds, i.e., cysteinylphenol (CP), cysteaminylphenol (CAP), and related compounds, that were synthesized by combining phenol or catechol with cysteine or cysteamine [10,23]. Phenolic compounds are known to be more selectively toxic to melanoma cells due to the specific activation of phenols by tyrosinase and the systemic effect of catechols by autoxidation [3]. Phenolic compounds are known to be more potent melanocyte-toxic agents than catechols in the in vivo system. The phenolic compound 4-*S*-cysteaminylphenol (4SCAP), an amine derivative of a sulfur homologue of tyrosine, was found to be selectively incorporated into melanoma tissues by whole-body autoradiography [16,29]. Thus, it was shown that 4*S*CAP has selective melanocyte toxicity and inhibits the growth of murine melanoma tissue. Furthermore, it is known that 4SCAP is a substrate for mammalian tyrosinase and is oxidized to form an *o*-quinone derivative [16].

Magnetic nanoparticles generate heat under an alternating magnetic field (AMF), and magnetic hyperthermia is an effective method to induce tumor cell death. Recent studies by many research groups have revealed that tumor immune mechanisms occur in vivo via heat-shock proteins (HSPs). In hyperthermia, the expression of HSPs plays an important role in the immune response [31,32,33,34,35,36,37,38]. Accumulating evidence from our group [39,40,41,42] indicated that hyperthermia-induced HSP expression is involved in tumor immunity, and hyperthermia with resultant immune induction (thermo-immunotherapy) may open a door to cancer treatment [43]. These results suggest that magnetic hyperthermia is applicable to the treatment of malignant melanoma. Taking advantage of the cell biological characteristics that melanoma cells are vulnerable to heat shock and are most susceptible to immune reactions, CAP can be immobilized on magnetite particles that generate cell-killing heat in response to magnetic field irradiation, thereby creating a novel targeted therapeutic approach, chemo-thermo-immunotherapy (CTI therapy).

To establish a specific melanoma-targeted treatment that has not been considered in the conventional concept, sulfur-containing tyrosine derivatives such as 4SCAP can be utilized to develop a nano-drug delivery system (DDS) and a melanoma-targeted systemic therapy. We chemically modified tyrosine, a substrate of the melanogenic enzyme tyrosinase, with thioether and synthesized a number of derivatives (Table 1) [9,10,15]. Among them is the *N*-propionyl derivative of 4SCAP, *N*-propionyl-4-*S*-CAP (NPrCAP), which is a sulfur derivative of the low-molecular-weight amino acid tyrosine, and a stable drug that was completely harmless to the skin and other organs when applied to animals intraperitonially, showing no systemic adverse effects except for the depigmentation of body coat [15,18]. NPrCAP showed a high substrate affinity for tyrosinase and exhibited melanoma-specific and irreversible cytotoxicity. This compound was found to possess both cytostatic and cytocidal effects on melanoma in vivo and in vitro through the *o*-quinone of NPrCAP [19,20]. We provided new evidence that CTI therapy can be developed by combining NPrCAP with magnetic nanoparticles, exploiting the unique melanogenesis cascade. For this purpose, we developed three nanoparticles: (1) NPrCAP encapsulated in magnetite cationic liposomes (NPrCAP/MCL) [44], (2) direct conjugation of magnetite nanoparticles with NPrCAP (NPrCAP/M) [45,46,47,48], and (3) conjugation of NPrCAP with magnetite nanoparticles via polyethylene glycol (NPrCAP/PEG/M). We first prepared NPrCAP/MCL to test the combined effects of chemotherapy using NPrCAP and magnetic hyperthermia on melanoma. An in vitro experiment showed that NPrCAP in NPrCAP/MCL had a dose-dependent effect on B16 cell proliferation, and the combination treatment of NPrCAP with magnetic hyperthermia was determined to have an additive effect. Next, we prepared NPrCAP/M by the direct conjugation of magnetite nanoparticles with NPrCAP for melanoma-specific DDS. Most of our animal experiments were carried out by utilizing NPrCAP/M. We then synthesized NPrCAP/PEG/M with long chains, expecting an increased cell binding for clinical human studies [46,48]. The detailed differences between NPrCAP/M and NPrCAP/PEG/M are described in Section 3.2 (b).

Our in vivo animal studies indicated that these agents first enter the endosomal system via receptors on the surface of the melanoma, then directly enter the metabolic pathway of melanogenesis and accumulate in the pathway of intracellular melanogenesis. We found that NPrCAP-conjugated magnetite nanoparticles selectively accumulate in melanoma tissues in vivo when administered locally or systemically, as well as in in vitro cultured cells. Specifically, when administered intraperitoneally, NPrCAP/M accumulated in melanosomes, which contain the final products of melanogenesis and accumulate intracellularly in melanoma [45] (Figure 1).

The CTI approach has two basic strategies. One is the drug-delivery system and the other is the generation of cytotoxic free radicals. Both are based on tyrosinase-mediated melanogenesis. The NPrCAP/magnetite complex has two cytotoxic phases of the cell destruction/death process. One is cell apoptosis resulting from oxidative stress exposure to tyrosinase, and the other is cell necrosis resulting from heat shock due to exposure to alternating magnetic field (AMF). *N*-propionyl-4-*S*-cysteaminyl-1,2-benzoquinone (NPrCAQ) reacts rapidly with R-SH (cysteine, GSH and bovine serum albumin) through sulfhydryl group to produce RS-*N*-propionyl-4-*S*-cysteaminylcatechol (NPrCAC). NPrCAQ and RS-NPrCAC upon autoxidation produce reactive oxygen species (ROS) such as hydrogen peroxide and superoxide radical, eliciting apoptosis of melanoma cells to lead cell death. Tyrosinase, which binds both AP (activator protein)-1 and -3, is transported to stage III melanosomes from tubular regions of early recycling endosome.

We have developed a unique novel DDS for clinical trials because NPrCAP was the most effective tyrosine derivative against melanoma preclinically and is an ideal drug as a DDS [48]. It is a substrate of melanogenesis with a much lower Vmax value and a higher Km value than tyrosine. It is selectively and irreversibly accumulated within the melanosomal compartment in melanocytes and melanoma cells via a mechanism considered to be based on a cell membrane receptor [46]. However, NPrCAP is insoluble in water. In order to develop a water-soluble NPrCAP, we synthesized a complex with dextran, a polysaccharide that induces extremely high solubility in water and low viscosity [48].

## 2. Advances in Anti-Melanoma Targeted Small Molecules and Mechanisms in Relation to the Melanogenesis Cascade

### 2.1. Chemotherapeutic Approaches Using the Initial Step of Melanogenesis: Tyrosine, Dopa and Their Analogues

There are two major categories of tyrosinase-interacting melanin precursors in the initial step of melanin biosynthesis that were examined as potential sources when designing antimelanoma agents. They were tyrosine (monohydroxy = phenolic) and dopa (L-dopa, levodopa) (dihydroxy = catecholic) [2]. 5,6-Dihydroxyindole is another melanogenesis-related group that has been explored for developing tyrosinase-mediated antimelanoma agents [49]. It is, however, extremely reactive, producing reactive oxygen species (ROS), and this reactivity has limited dihydroxyindoles’ role as antitumor agents.

(a)Tyrosine

The degree of melanin pigmentation of in vitro melanoma cells can be altered by changing the concentration of tyrosine in the culture medium [50]. Raising the concentration of tyrosine causes an increase in vitro melanoma cell growth. Tyrosine levels may be elevated enough to cause toxicity to melanin-forming cells [51]. This technique has proved to be a biologically useful method. A major obstacle to applying this approach for melanoma therapy is, however, that tyrosine is one of the naturally occurring amino acids and is extensively used in general protein synthesis. Rapidly growing melanocytes incorporate only about 5% of exogenous tyrosine into melanin biosynthesis, with the remainder entering protein synthesis. Therapeutic approaches to raise or lower levels of tyrosine in vivo have, therefore, had limited success.

(b) L-Dopa/dopamine and related analogues

L-dopa is an amino acid, but it is not normally found in cellular proteins. L-dopa may, therefore, be selectively incorporated into melanoma cells. Through the use of L-dopa decarboxylase inhibitors, which limit the decarboxylation of L-dopa to dopamine, an enhanced incorporation of L-dopa into melanoma tissues was achieved in vivo [52]. L-dopa was selectively cytotoxic to melanoma cells in vitro [53], and through the use of its more water-soluble analog L-dopa methyl ester, in vivo antimelanoma activity was observed in the experimental murine melanoma [54]. In this animal experiment, a hypercatecholamine-type state was observed [54]. The animals became agitated and tremulous and usually died within 1–2 h after administration. These acute toxic effects of L-dopa are probably mediated by its conversion to dopamine by the enzyme, dopadecarboxylase. Furthermore, the major metabolite of L-dopa, dopamine, was observed to be a highly potent inhibitor of melanoma cells in vivo and in vitro [55]. Interestingly, however, four patients with extensive melanoma metastases were treated by dopamine infusions at maximal tolerable levels for from 48 to 120 h and labeling and scintillation indices of the tumors were measured. A comparison of the pre- and post-infusion data showed a significant reduction in the % labelling index, from 1.0~3.0 to 0.1~0.2, indicating a consistent 10-fold reduction. However, persistent fatigue precluded further retreatment [56] (Table 2).

One of the L-dopa analogues, 6-hydroxydopa, was found to be highly potent in terms of selective interaction with tyrosinase and toxicity for melanoma cells [49,57]. However, in in vivo experiments, 6-hydroxydopa was found to exhibit unique neurotoxic properties that resulted in the selective degeneration of adrenergic nerves [2].

Following those efforts by Wick’s group, Fujita et al. reported that 5-*S*-cysteinyldopa (5SCD), a pheomelanin precursor, was selectively toxic to melanoma cells in vitro and in vivo [58]. This study showed that 1 mM 5SCD exhibited a similar or an even higher degree of growth inhibition to 6 mM L-dopa. In an extension of this study, Ito et al. synthesized several new molecules that are structurally related to 5SCD and demonstrated melanoma activity in vitro and in vivo [9]. However, the use of catecholic melanin precursors such as L-dopa and 5SCD faced a drawback in that they tend to undergo autoxidation [59] leading to the production of reactive oxygen species such as hydrogen peroxide [60], resulting in non-selective cytotoxicity [4,9].

(c) Sulfur homologues of tyrosine and related compounds

The use of phenolic melanin precursors appeared more rational and promising for the development of melanogenesis-based anti-melanoma agents because the oxidation of phenolic compounds is only dependent on tyrosinase present in melanocytes and melanoma cells. To examine this possibility, 4-*S*-cysteinylphenol (4SCP), a phenolic thioether, was synthetized by heating phenol and cystine in hydrobromic acid, and its anti-melanoma effects were evaluated [9] (Table 1). 4SCP is the sulfur homolog of L-tyrosine, the natural melanin precursor, and was expected to be a good substrate for tyrosinase. In fact, 4SCP was found to be as good a substrate for melanoma tyrosinase as L-tyrosine [16] and a good inhibitor of L-tyrosine transport to melanoma cells, implying the efficient transportation of 4SCP [61], and ^3^H-4SCP was specifically incorporated into cultured human melanoma cells in vitro [62] and mouse melanoma tissues in vivo [16]. However, the antimelanoma effects of 4SCP were found to be minimal [16,21], which led us to modify its structure to the amine analog 4-*S*-cysteaminylphenol (4SCAP), in the hope that this modification would lead to the more effective incorporation into melanoma cells and oxidation by tyrosinase to elicit greater antimelanoma effects. 4SCAP was readily prepared by heating phenol and cystamine in hydrobromic acid [10]. In fact, 4SCAP was found to cause a significant inhibition of in vivo melanoma growth and marked depigmentation of black skin and hair follicles, with the effects being much greater than those caused by 4SCP [10,23,25,28,29]. When hair follicles were plucked from adult black mice to stimulate new melanocyte growth and activate tyrosinase synthesis, subsequent repeated intraperitoneal injection (ip) administration of 4SCAP resulted in 100% new growth of white hair follicles at the site at which black follicles were plucked [23]. 4SCAP also suppressed lung colony formation after intravenous inoculation of B16 melanoma cells in mice [25,29]. These promising in vivo results were paralleled by in vitro studies showing that 4SCAP was much more toxic to cultured human melanoma cells than 4SCP [21] and more potent in inhibiting protein synthesis [16].

However, there were certain drawbacks to further exploiting 4SCAP as an antimelanoma agent: the maximal tolerable dose was limited due to its low water solubility, low LD_50_ value [24], and hypotensive effect [14]. 4SCAP acts as a substrate for a number of enzymes in addition to tyrosinase, including dopamine β-hydroxylase [14] and monoamine oxidase [17], which catalyze production of the sulfoxide and aldehyde, respectively. As a result, 4SCAP appeared to cause adverse effects when administrated systematically. Recent progress in nanotechnology may overcome the limitations, such as low solubility and adverse effects of compounds, and exploring the nanoformulation of 4SCAP is a possible approach [63]. Alternatively, we took the approach of modifying 4SCAP by chemical synthesis. 4SCAP homologues such as α-methyl-4SCAP, 4-*S*-homocysteaminylphenol, and *N*,*N*-dimethyl-4SCAP were also prepared, and their depigmenting effects were examined in vitro [11] and in vivo [29]. However, none were found to be superior to 4SCAP. Attempts to lower the hypotensive effect of 4SCAP and increase the efficacy of tyrosinase-dependent cytotoxicity were made using enantiomers of α-methyl-4SCAP and α-ethyl-4SCAP [12,26]. However, improvements were limited. Another attempt to increase the efficacy of 4SCAP as an antimelanoma agent, using the catecholic derivative 4-*S*-cysteaminylcatechol (4SCAC), had limited success due to systemic toxicity [13]. To overcome these difficulties, we prepared *N*-acetyl-4SCAP (NAcCAP) by reacting 4-hydroxythiophenol with 2-methyl-2-oxazoline using the method of Padgette et al. (1984) [14].

NAcCAP was found to act as a substrate for mushroom tyrosinase as well as 4SCAP, and exhibit greater in vivo antimelanoma activity than 4SCAP, although at higher doses [24]. Importantly, NAcCAP showed marked water solubility [24] and a greater depigmenting effect on follicular melanocytes than 4SCAP [29]. A single ip administration of NAcCAP into newborn mice led to the development of silver hair follicles in the entire body. The selective destruction of melanocytes was observed at 12 hr after a single ip injection. None of the surrounding keratinocytes or fibroblasts showed such subcellular degeneration and cell death [30]. ^14^C-NAcCAP was specifically taken up by melanotic melanoma cells, but not by amelanotic melanoma cells [19].

These promising results that were achieved by protecting the amino group in 4SCAP were elaborated further by replacing the *N*-acetyl group with an *N*-propionyl group. NPrCAP was readily prepared by reacting 4-hydroxythiophenol with 2-ethyl-2-oxazoline [15]. NPrCAP was expected to exhibit greater uptake into melanoma cells because of its greater lipophilicity compared with NAcCAP. NPrCAP was found to have a greater Vmax value for tyrosinase than tyrosine, 4SCAP, and NAcCAP had, and caused marked depigmentation of black hair follicles in adult and newborn C57 mice, with biochemical and morphologic features indicative of apoptosis [15,18]. The tyrosinase-mediated cytotoxicity of NPrCAP was further confirmed by the finding of decreased viability of tyrosinase-transfected COS7 monkey-kidney cells expressing high tyrosinase activity. NPrCAP, however, also transiently inhibited the proliferation of a tyrosinase-negative albino melanocyte line, as well as COS7 cells [18]. It appears that the major process of NPrCAP melanocyte-toxicity involves cytocidal apoptosis associated with active tyrosinase. In addition, there is transient, non-tyrosinase-mediated cytostatic cytotoxicity. Antimelanoma effects were also compared for NPrCAP and NAcCAP in B16 melanoma tumors; the two compounds were comparable in terms of their growth-inhibitory effect, but NPrCAP was considerably better than NAcCAP in increasing the lifespan of melanoma-bearing mice [15]. Taken together, NPrCAP appears to be the best antimelanoma agent among the sulfur-containing tyrosine analogs (phenolic thioethers) developed to date.

A number of attempts to increase the efficacy of NAcCAP have been reported. Robins et al. [64,65,66,67,68] synthesized various NAcCAP analogues with the intention of increasing the lipophilicity of the compounds. A modest increase in antimelanoma activity against several melanoma cell lines was observed, which was correlated with increased lipophilicity. However, those compounds also exhibited tyrosinase-independent cytotoxicity against an amelanotic SK-Mel-24 melanoma and an ovarian cell line. Of particular interest is a recent report showing a hybrid of 4SCAP with triazene, a DNA-alkylating compound [69]. Those hybrids were found to be excellent tyrosinase substrates. Some of those compounds were unexpectedly devoid of hepatotoxicity while maintaining cytotoxic activity in melanoma cells. 4SCAP appears to be an important component for the new strategy of developing anti-melanoma agents.

### 2.2. Mechanism of Anti-Melanoma Action in Relation to the Melanogenesis Cascade

Here, we describe the possible mechanisms of anti-melanoma action of 4SCAP, NAcCAP/NPrCAP, and related phenols. Tyrosinase is certainly a trigger of anti-melanoma effects of 4SCAP based on the results that (1) 2-*S*-cysteaminylphenol, an isomer of 4SCAP, did not act as a substrate of tyrosinase and did not exhibit any anti-melanoma and depigmenting effects [10], (2) the cytotoxicity of 4SCAP against various melanoma cell lines depended on the degree of their pigmentation [22], and (3) 4SCAP exhibited cytotoxicity to melanocytes in black mice but not in albino mice [23].

Tyrosinase is known to catalyze oxidation of various phenols to the corresponding *o*-quinones [70]. *o*-Quinones are highly reactive molecules through binding with various functional groups, especially sulfhydryl and amino groups [8,71]. The binding of *o*-quinones with the cellular small sulfhydryl compounds cysteine and glutathione (GSH) produces the cysteinyl- and glutathionyl-catechol derivatives. This process results in various biochemical consequences in melanocytes and melanoma cells (Figure 2). One is the production of pheomelanic pigments by the oxidation of cysteinyl-catechol conjugates, similar to the production of natural pheomelanin from L-tyrosine. Pheomelanins are known to exhibit potent pro-oxidant activities [72]. Another is the depletion of GSH in cells that exhibit tyrosinase activity. Both of these biochemical events might lead to tyrosinase-dependent cytotoxicity. GSH depletion was shown to play a key role in the depigmenting and melanocytic action of NAcCAP: the co-administration of *N*-acetylcysteine, which up-regulated GSH content, completely abolished the depigmenting potency of NAcCAP, whereas buthionine sulfoximine, which depleted the tissue GSH content, enhanced the depigmenting potency of NAcCAP [73]. The co-administration of buthionine sulfoximine also significantly enhanced the antimelanoma effects of NAcCAP [74]. Additionally, *o*-quinones are capable of binding with enzymes and other proteins through their cysteine residues [75]. The binding of *o*-quinones to sulfhydryl enzymes essential for proliferation, such as thymidylate synthase and DNA polymerase, leads to the inhibition of DNA synthesis and cell growth [76]. Interestingly, upon tyrosinase oxidation, 4SCAP was found to be five-fold more effective than 4SCP in binding to alcohol dehydrogenase, a sulfhydryl enzyme [16,77].

In addition to the cytotoxicity of *o*-quinone due to their reactivity with sulfhydryl compounds, there is another possible mechanism for *o*-quinone cytotoxicity (Figure 2). *o*-Quinones may undergo one-electron reduction with reducing agents such as NAD(P)H and Fe^2+^ to produce semi-quinone radicals, which disproportionately form the catechol and the *o*-quinone [78,79]. Catechols are converted back to *o*-quinones upon autoxidation, concomitantly producing ROS such as superoxide radical, hydrogen peroxide, and hydroxyl radical. *o*-Quinones may also undergo two-electron reduction, with reducing enzymes such as NAD(P)H quinone dehydrogenase 1 (NQO1) or reducing compounds such as ascorbic acid, to produce catechols. Thus, both catechols and *o*-quinones can be a source of ROS generation through redox cycling [78,79,80]. In addition, the reactions of *o*-quinones with biological small thiols such as cysteine and GSH produce catechol derivatives that may produce ROS (Figure 1). Interestingly, catechols with cysteine conjugation, such as 5SCD, are more cytotoxic to melanocytes through the production of hydrogen peroxide than are the parent catechols such as L-dopa [58,60].

Whether a catechol or its corresponding *o*-quinone structure is more cytotoxic depends on the reactivity of the catechol: more reactive and, thus, cytotoxic catechols such as 6-hydroxydopamine become less reactive when oxidized to the *o*-quinone structure [59]. On the other hand, the sulfhydryl reactivity of *o*-quinone oxidation products contributes to the cytotoxicity of dopamine and *N*-acetyldopamine [59]. In our case, which of the two mechanisms prevails, the sulfhydryl reaction or the ROS production, is not known at present.

A unique feature of 4SCAP is that, upon tyrosinase-catalyzed oxidation, *o*-quinone derived from 4SCAP undergoes a facile cyclization through the amino group to form a reactive, cyclic quinone, dihydro-1,4-benzothiazine-6,7-dione [81,82]. This cyclic quinone was found to be highly cytotoxic to B16 melanoma cells in vitro and in vivo, much more so than 4SCAP [82].

Lastly, we examined how extensively ROS are involved in the tyrosinase-dependent antimelanoma and depigmenting effects of 4SCAP and NAcCAP (or NPrCAP). The production of ROS from the phenolic tyrosinase substrate NPrCAP in melanoma cells was reported by [27]. This study showed that the growth suppression of pigmented melanoma cells by NPrCAP was associated with an increase in intracellular ROS, activation of caspase 3, and DNA fragmentation. This study also showed that intratumoral administration of NPrCAP suppressed the growth in not only primary B16F1 melanoma tumors but also secondary, re-challenged tumors. The participation of CD8^+^ T cells was suggested for the NPrCAP-mediated anti-B16 melanoma immunity. In a following study, the molecular mechanisms of the NPrCAP cytotoxicity and immunogenicity were examined [20]. NPrCAP was shown to be oxidized by mushroom tyrosinase to the *o*-quinone *N*-propionyl-4-*S*-cysteaminyl-1,2-benzoquinone (NPrCAQ). NPrCAQ rapidly reacted with biologically relevant thiols, cysteine, GSH, and bovine serum albumin through the sulfhydryl group (Figure 2). The production and excretion of NPrCAQ-protein adducts was confirmed in B16F1 melanoma cells in vitro and in B16F1 melanoma-bearing mice in vivo. The protein fraction was hydrolyzed by heating in 6 M HCl and the amino acid released from the protein adduct, 5-*S*-cysteaminyl-3-*S*-cysteinylcatechol, was identified by HPLC. These results suggest that tyrosinase in melanoma cells activate the phenolic NPrCAP, acting as a prohapten, to the quinone-hapten NPrCAQ, which binds to melanosomal proteins to form possible neo-antigens, thus triggering an immunological response.

## 3. Thermal Medicine for Selective Anti-Melanoma Therapy Utilizing Melanogenesis Small Molecules

In recent years, two major melanogenesis-targeted anti-melanoma thermal medicine approached have been introduced with some clinical trials in Japan.

### 3.1. Melanogenesis Molecule-Based Boron Neutron Thermal Medicine: Reaction of Thermal Neutrons with Boron 10 Conjugated to Dopa Analogue Para-Boronophenylalanine Hydrochloride (^10^B_1_-BPA)

Boron neutron thermal medicine is based on the nuclear reaction between boron-10 (^10^B_1_) and thermal neutrons. The successful treatment of melanoma with boron neutron thermal medicine depends mainly upon melanogenesis molecule-based accumulation of ^10^B_1_ in the targeted tumor.

(a)Principle and pharmacokinetics of ^10^B_1_-BPA Thermal Neutron Capture therapy

The absorption of thermal neutrons, which are generated by an atomic reactor, from the nonradioactive isotope boron-10 (^10^B_1_) results in the emission of α particles and lithium atoms (^10^B(n,α)^7^Li reaction). The traveling range of the charged particles is measured to be 10–14 μm from the point of the neutron-activated boron atom. This is approximately equal to the diameter of a single melanoma cell. It was thus expected that, if ^10^B_1_ can be concentrated in melanoma cells, all the melanoma cells in a tumor can be destroyed without seriously injuring the surrounding normal tissues. The melanogenesis substrate, dopa was utilized for this purpose. It was expected that conjugating dopa with ^10^B_1_ would empower a selective, biological, nonsurgical treatment of melanoma [83]. Four hybrid compounds of ^10^B_1_ and dopa were synthesized, and ^10^B_1_-*para*-boronophenylalanine hydrochloride (^10^B_1_-BPA) was found to be the most promising [84] (Table 3).

When boron-labeled dopa was administrated, the boron uptake was found to be proportional to the rate of melanin synthesis. The specific affinity and binding of ^10^B_1_-BPA for melanoma was examined by density gradient fractionation of melanoma cells and subsequent isolation of the melanosome fraction. The subcellular fractionation of melanosomes revealed a greater concentration of ^10^B_1_ than any other subcellular fractions [85]. In parallel to measuring this chemical and physical ^10^B_1_ distribution in melanoma tissue, comparative visualization was carried out in the whole body using α-track autoradiography. A high concentration of ^10^B_1_ in melanoma was observed at 30 min after administration, showing as a bright region in the α-track autoradiography [86].

(b) In vivo radiotherapeutic studies and preclinical experiments

The in vivo therapeutic effects of ^10^B_1_-BPA thermal neutron capture therapy tested on experimental animal systems were found to have a previously unobserved melanoma-killing effect [84,87]. A number of preclinical therapeutic experiments were carried out using experimental animal models such as melanoma-bearing hamsters and pigs, and human melanoma-bearing nude mice. In addition, the distribution of radiation energy to which the whole body is exposed following a local exposure was assessed using a human phantom [88]. These experiments showed the selective accumulation of ^10^B_1_-BPA by the melanoma cells, causing the greatest amount of absorbed radiation energy to be limited to the target tissue.

(c) Clinical trial of melanoma patients using ^10^B_1_-BPA thermal neutron capture therapy (Boron Neutron Capture Therapy, BNCT)

The first patient in a preliminary clinical trial was a 66-year-old male with an inoperable malignant melanoma lesion on the left occiput. At this stage, the patient suffered from persistent double vision, nausea, and headaches. The irradiation was carried out with a dosage of 1 × 10^13^ n/cm^2^ at the melanoma surface in the same position for 2 h and 19 min. Two months after a single application of BNCT, marked regression of the tumor was observed with improvements in double vision, nausea, and headaches. Furthermore, regression of the treated melanoma without any sign of regrowth was seen more than 9 months after application of the melanoma-targeted BNCT, strongly indicating a cure of the treated lesion, and the symptoms of double vision, nausea, and headaches disappeared completely [89]. Subsequently, 22 melanoma patients suffering from various stages of malignant melanoma were treated with this ^10^B_1_-BPA BNCT [90,91] (Table 4).

Despite the remarkable achievements of melanoma-targeted BNCT, there appear to be several limitations that may inhibit the development of this therapeutic system as a novel melanoma therapy for humans. The limitations may include the clinical safety, therapeutic efficacy, and specificity of ^10^B_1_-BPA BNCT. In addition, amelanotic melanoma exhibited relatively lower ^10^B_1_-BPA accumulation than high-melanin-producing melanoma [95,96]. Another obstacle is the limited availability of irradiation facilities equipped with thermal columns and a source of thermal neutrons.

### 3.2. Melanogenesis-Based Antimelanoma Thermal Medicine by Conjugation with Magnetite Nanoparticles; Establishment of Melanoma Chemo-Thermo-Immunotherapy (CTI Therapy)

(a)Principle of magnetite hyperthermia for cancer nanomedicine

Recent progress in nanotechnology has led to the creation of novel cancer treatments. Nanoparticles such as magnetic nanoparticles and gold nanoparticles can generate heat upon irradiation with an alternating magnetic field (AMF; 100 kHz–400 kHz) for magnetic hyperthermia and with a near-infrared laser (700 nm–900 nm) for photothermal therapy, respectively (Figure 3). Nanoparticle-mediated thermotherapy enables the specific heating of tumors where the nanoparticles are delivered by DDS. As compared with photothermal therapy, magnetic hyperthermia using magnetic nanoparticles can be applied to relatively deep tumors, owing to the excellent tissue permeability of magnetic fields. Among the magnetic nanoparticles, magnetite (Fe_3_O_4_) nanoparticles (diameter 10–100 nm) are considered promising nano-heaters in the body due to their high stability and low toxicity [97]. For the DDS of magnetite nanoparticles, surface modifications of the nanoparticles with polymers such as dextran [98] and PEG [99] have been constructed for in vivo experiments. We focused on the melanoma targeting of NPrCAP and synthesized two types of NPrCAP-conjugated magnetite nanoparticles, NPrCAP/M and NPrCAP/PEG/M, for animal and human studies, respectively [45,48].

(b) Conjugation of melanogenesis molecule NPrCAP with magnetite nanoparticles for novel antimelanoma thermotherapy

Magnetite nanoparticles (for NPrCAP/M in animal studies) and dextran magnetite (for NPrCAP/PEG/M in human studies) were coated with an aminosilane (Figure 4). NPrCAP was bound on the surface of the aminosilane-coated magnetite nanoparticles to synthesize NPrCAP/M. For NPrCAP/PEG/M synthesis, the aminosilane-coated dextran magnetite was reacted with PEG-NPrCAP. NPrCAP/PEG/M are chemically stable and can be produced in large quantities for human studies. Both NPrCAP/M (in animal studies) and NPrCAP/PEG/M (in human studies) employed to effect melanoma-targeted chemotherapy (by the NPrCAP component), thermotherapy (by the magnetite nanoparticles), and resultant immunotherapy, providing a basis for a novel CTI therapy [46].

Transmission electron microscope observation [45,100] revealed the selective accumulation of NPrCAP/M in melanoma cells by active transport through a still-unknown receptor system (Figure 5) [62]. Specifically, NPrCAP/M nanoparticles were found to be incorporated into early- and late-stage melanosomes, to which tyrosinase is also transported from the trans-Golgi network (TGN). Once NPrCAP/M is incorporated into melanosomes in melanoma cells, it would be retained and accumulate, as few melanosomes are transferred from melanoma cells. Thus, it is expected that heat generation from NPrCAP/magnetite nanoparticles exposed to AMF will yield the selective disintegration of melanoma cells. [97].

(c) Development of the chemo-thermo-immuno-therapy approach

The AMF exposure of B16F1 melanoma-bearing mice treated with NPrCAP/M yielded the selective disintegration of melanoma tissues, as shown by hematoxylin-eosin [101] (Figure 6) and Berlin blue staining [45]. Importantly, NPrCAP/M treatment with and without AMF yielded almost identical degrees of the growth inhibition of primary transplants (Figure 7a–c), indicating that NPrCAP/M alone had a significant chemotherapeutic effect. However, a marked difference in melanoma growth between those two groups appeared after the primary tumors were removed and the mice were re-challenged with a second melanoma transplantation without further treatment. Magnetic hyperthermia using NPrCAP/M resulted in the most significant growth inhibition of the re-challenge melanoma (i.e., 30–50% complete rejection of re-challenge melanoma growth) and increased the lifespan, indicating that magnetic hyperthermia using NPrCAP/M has a thermo-immunotherapeutic effect (Figure 7a,b,d,e and Figure 8a,c) [102]. Further investigation showed that the thermo-immunotherapy against re-challenge B16F1 melanoma was more effective at a temperature of 43 °C for 30 min, repeated once every other day three times, than at 46 °C (Figure 8a–c).

In 2003, Ito et al. reported that magnetic hyperthermia induces antitumor immunity from the release of HSP70-peptide complexes during necrotic tumor cell death [103]. HSP70 production was then analyzed in primary melanoma tumors undergoing CTI therapy, and CD4^+^ and CD8^+^ T cell infiltration was studied in re-challenge secondary tumors [102]. NPrCAP/M-mediated, AMF-induced hyperthermia at 43 °C for 30 min and at 46 °C for 15 or 30 min produced similar increases in HSP70 level (Figure 8a,d). Although thermotherapy could induce HSP70 as abundantly at 46 °C for 15 min as at 43 °C for 30 min, the former condition failed to suppress the re-challenge melanoma transplant as well as 43 °C thermotherapy (Figure 8b,c). This suggests that some immunological factors other than HSPs may be partly responsible for the rejection of re-challenge melanomas. Hyperthermia at 43 °C for 1 h resulted in augmentation of MHC class I, associated with induced expression of HSP70 [104]. Hyperthermia treatment of tumor cells enhances cross-priming, possibly via the up-regulation of HSPs [43]. Thus, it may be postulated that, by inducing HSPs, magnetic hyperthermia using NPrCAP/M at 43 °C could be an effective therapeutic modality for metastatic melanoma. It is further speculated that NPrCAP plays two important roles in anti-melanoma approaches, i.e., the selective incorporation of nanoparticles into melanoma cells and unique induction of chemo-thermo-immunotherapy effects on melanoma cells.

(d) Immunological effects of melanogenesis-targeted anti-melanoma molecules; orchestration of innate and adaptive immunity by CTI therapy

As described above, we observed that the antitumor effect of NPrCAP/M with AMF exposure was superior to that of NPrCAP/M alone [102,105]. The treatment of primary melanoma with NPrCAP/M plus AMF exposure showed a significant growth inhibition for untreated re-challenge melanoma and increased the lifespan and survival rate of the host animals, i.e., 30–50% complete rejections of re-challenge melanoma growth (Figure 7e and Figure 8c) whereas NPrCAP/M alone yielded no complete rejections (Figure 7e), indicating that NPrCAP/M with AMF exposure produces a strong immunotherapeutic effect. Since we had already shown that heat-induced HSP70 plays an important role in the induction of melanoma-specific immunity, we further investigated the role of HSPs in the induction of antitumor immunity by NPrCAP/M with AMF exposure. HSPs are classified in families such as HSP40, HSP70, HSP90, HSP100, and small HSPs and are uniquely upregulated by several kinds of stress such as heat, arsenite, hypoxia, low glucose, etc. Therefore, we examined which HSPs were upregulated by our CTI therapy in B16F1 and B16-OVA melanoma cells in culture [101]. We observed that HSP70, HSP90, and ER-resident HSPs, including gp96, were upregulated. Then, we investigated which HSP was most responsible for the induced tumor immunity by NPrCAP/M with AMF exposure. A depletion assay using each anti-HSP antibody showed that, although HSP70, HSP90, and gp96 were all shown to bind melanoma antigen peptides such as a TRP-2-derived peptide, HSP70 was shown to be largely responsible for the anti-melanoma immunity. In our study, HSP70 showed the highest upregulation among the HSPs in response to NPrCAP/M with AMF exposure; newly generated HSP70 may have more chances to bind melanoma-associated antigen peptides. Melanoma cells treated with NPrCAP/M plus AMF exposure underwent necrotic cell death; then, HSP70, HSP90, and gp96 were released into the extracellular milieu and were taken up by dendritic cells (DCs). These HSPs participated in the cross-presentation of melanoma-associated antigen peptides (OVA peptide and TRP-2 peptide) by the DCs to specific CD8^+^ T cells through chaperoning the peptides. CD91 serves as an HSP receptor expressed on DCs for the cross-presentation of antigen peptides chaperoned by HSPs [106,107].

In addition to the activation of melanoma-specific immune responses (adaptive immunity) via production of HSP-peptide complexes mentioned above, HSPs are well-known to act as danger signals that can activate innate immunity (Figure 9). HSP60, HSP70, HSP90, and gp96 have been demonstrated to stimulate Toll-like receptor 4 (TLR4) to promote production of pro-inflammatory cytokines such as IL-1β, IL-6, TNF-α, and IL-12 through the NF-kB pathway [108,109]. Our production of hyperthermia using NPrCAP/M with AMF exposure induced necrotic melanoma cell death, resulting in the passive release of various types of HSPs into the extracellular milieu. As a result, released HSPs induced the activation of innate immunity via binding to TLR4 or other HSP receptors expressed on infiltrated antigen-presenting cells such as dendritic cells and macrophages. Thus, CTI therapy orchestrated both innate and adaptive immunity, thereby eliciting strong anti-melanoma immunity.

The melanoma cells undergo necrotic cell death after CTI therapy. As a result, the HSP-peptide complex, which is increased by heat stress, is released and taken up by antigen-presenting cells such as dendritic cells (DCs). HSP can stimulate the innate immunity through binding to Toll-like receptors (TLRs) and also activate the adaptive immunity via cross-presentation of melanoma antigen peptides chaperoned by HSPs to specific CD8^+^ T cells.

Furthermore, since CTI therapy elicits systemic anti-melanoma immunity, it can be a promising therapy for the prevention of recurrence and/or distant metastasis of melanoma. To prove this, we examined whether CTI therapy of primary cutaneous B16 melanoma can inhibit colonization in lungs by intravenously injected secondary, re-challenge B16 melanoma cells. We observed that the treatment of primary cutaneous B16 melanoma with NPrCAP/M plus AMF exposure (CTI therapy) clearly inhibited lung metastasis formation compared with treatment with NPrCAP/M alone. These results demonstrated that CTI therapy induced systemic anti-melanoma immunity, and therefore prevented lung metastasis and the recurrence of melanoma.

Thus, CTI therapy against advanced melanoma is a useful strategy, not only for the treatment of primary melanoma, but also for preventing the recurrence and metastasis of melanoma.

(e) Preliminary clinical trials for human melanoma patients

The results of preliminary clinical trials which were reported previously [46,48]. According to these reports, the trials were conducted with stage III and IV melanoma patients (approved by Clinical Trial Research Protocol No. 18-67, Sapporo Medical University with funding for research on Advanced Medical Technology from the Ministry of Health, Labor and Welfare of Japan to KJ as the Principal Investigator, [Project No. H21-Nano-6]). The study was carried out by utilizing NPrCAP/PEG/M, to improve dispersion stability for administration. Four patients were evaluated and none showed any significant difficulties after repeated treatment. In addition, one showed a complete response, and one had a partial response, and both of these patients were able to carry out normal daily activities for more than 32 months after CTI therapy. It is expected to carry out large-scale clinical trials to evaluate the overall therapeutic effect and define the molecular interactions between the chemotherapeutic and thermo-immunotherapeutic effects.

## 4. Anti-Melanoma Approach for Unresectable/Metastatic Melanoma Patients by Currently Available Targeted Therapies and Immune Checkpoint Inhibitors

Until the first decade of the 21st century, treatment for unresectable/metastatic melanoma had long been largely ineffective. The only chemotherapeutic drug shown to have efficacy for melanoma is dacarbazine, but its overall response rate and 5-year overall survival (OS) were about 10% and less than 10%, respectively [110]. There were no combination chemotherapies that had superior efficacy to dacarbazine monotherapy. However, since 2011, new types of therapies/drugs, i.e., immune checkpoint inhibitors (ICI), BRAF inhibitors (BRAFi), and MEK inhibitors (MEKi), have entered the market and revolutionized the therapeutic strategies for unresectable/metastatic melanoma. Moreover, some of the drugs are used in an adjuvant setting and prolong recurrence-free survival (RFS) and/or OS of patients with resected stage III melanoma. It was recently reported that the ICI pembrolizumab significantly reduced the risk of disease recurrence or death versus placebo as an adjuvant treatment in patients with stage IIB/C melanoma [111].

Neoplastic cells expand their population by escaping from the host immunosurveillance. One of the mechanisms for escaping is the exploitation of immune checkpoints in antigen presenting cells (APC) and T lymphocytes [112]. Immune checkpoints essentially inhibit autoimmunity and exaggerate immune responses to protect the host tissues, but they also inhibit cancer immunity by causing an exhaustion of anti-cancer immune cells and thereby help the survival and proliferation of neoplastic cells. ICI are antibody drugs that target the immune checkpoint molecules and reactivate cancer immunity. Ipilimumab was approved by U.S. Food and Drug Administration in 2011 and nivolumab and pembrolizumab were approved in 2014. These drugs dramatically improved the treatment of melanoma. On the other hand, ICI frequently cause immune-related adverse events (irAE) [113].

### 4.1. Ipilimumab

Ipilimumab is an antibody drug that blocks cytotoxic T lymphocyte-associated 4 (CTLA-4; CD152). CTLA-4, a protein in the CD28 family of costimulatory molecules, is a cell surface receptor expressed in T lymphocytes. A cytotoxic T lymphocyte (CTL) becomes activated when CD28 on its surface binds to CD80/86 on the surface of an APC. However, when CTLA-4 on a CTL binds to CD80/CD86 on an APC, it inhibits CTL activity. Ipilimumab cancels the “brake” on an anti-neoplasm response by blocking CTLA-4. Ipilimumab, for the first time, improved the OS of patients with untreated metastatic melanoma in a setting of a combination therapy with dacarbazine. In a phase III trial randomized to ipilimumab plus dacarbazine vs. dacarbazine alone, the median OS of each arm was 11.2 vs. 9.1 months, respectively [114]. This was a monumental work of immunotherapy; not only for melanoma, but for all cancer types. In addition, tail-of-the-curve benefits were observed in patients treated with ipilimumab, which means that some of the patients who responded experienced a long-term survival well beyond the median. The overall incidences of all-grade irAE and grades 3/4 irAE were 72 and 24%, respectively [115]. Since it later became clear that PD-1 blockers are superior to ipilimumab in terms of efficacy and safety, ipilimumab is now used in a combination therapy with nivolumab or in a second line monotherapy.

### 4.2. Anti-PD-1 Antibodies: Nivolumab and Pembrolizumab

Programmed cell death-1 (PD-1; CD279) is expressed on the cell surface of activated T lymphocytes. When PD-1 is bound by PD-L1 or PD-L2 expressed on the cell surface of neoplastic cells or APC, the T lymphocytes become inactivated. Nivolumab and pembrolizumab block PD-1 and reactivate cancer immunity. In the 6.5-year outcome of the CheckMate-067 study, patients with untreated, unresectable melanoma were randomly assigned to nivolumab plus ipilimumab, nivolumab monotherapy, or ipilimumab monotherapy [116]. Median OS was 72.1, 36.9 and 19.9 months, respectively. The 6.5-year OS rates were 57%, 43% and 25%, respectively. As for pembrolizumab, the Keynote-001 study showed 5-year survival outcomes of patients with previously treated or treatment-naive advanced/metastatic melanoma [117]. The estimated 5-year OS rate was 34% among all patients and 41% among treatment-naive patients; median OS was 23.8 and 38.6 months, respectively. These studies showed a significantly improved OS with anti-PD-1 antibodies compared with ipilimumab. The CheckMate-067 study also showed the efficacy and safety of combination therapy with nivolumab and ipilimumab. The combination therapy showed increased progression-free survival compared with nivolumab monotherapy, although the OS difference did not reach statistical significance. The combination therapy is the most effective treatment at present. However, the combination therapy showed a significantly higher occurrence of irAE. The rate of grades 3/4 irAE were 59%, 23%, and 28% in the nivolumab plus ipilimumab, nivolumab, and ipilimumab arm, respectively [118].

## 5. Summary and Conclusions

Despite advances in early detection, malignant melanoma still causes high cancer-related mortality. In a subset of patients with metastatic melanoma, surgical resection is not possible, and systemic therapies are needed. Fortunately, over the last decade, novel systemic-targeted immunotherapeutics have been successfully introduced to improve both the quality of life and survival of high-risk patients. However, there are still a number of unsolved problems. There are still many patients who do not benefit from the current immunotherapies, and there are no reliable biomarkers that can predict the efficacy of immunotherapies. Furthermore, certain forms of cutaneous melanomas, such as acral plantar type and non-cutaneous forms such as mucosal and uveal melanomas, are frequently resistant to the current immunotherapies. These resistant melanomas are immunologically “cold” tumors, which express few cancer-antigens and escape from host surveillance for neoplasms. One of the potential approaches to augment the efficacy of immunotherapies is disruption of melanoma cells to release neoantigens to host immunity by a system such as our CTI therapy. In this context, many clinical trials of immunotherapies in combination therapies have recently been carried out, including with small molecule targeted drugs, cytotoxic chemotherapies, radiotherapies, and oncolytic viral therapies.

The development of melanogenesis-targeted drugs can, therefore, be a promising research area, because melanin synthesis is a unique biochemical pathway operating only in melanocytes and their neoplastic counterpart, melanoma cells. The key enzyme of this pathway is tyrosinase, which is always retained in melanoma cells in vivo, even when the whole tumor becomes non-pigmented and amelanotic. Melanin metabolites are always composed of small molecules that are synthesized by melanoma cells. The tumor can be easily targeted by small molecules that chemically mimic melanogenesis substrates. In addition, small molecules of melanin metabolites are toxic to melanocytes and melanoma cells and may kill them [119,120]. Products of the tyrosinase reaction (melanin intermediate molecules) can thus exhibit useful selective cytotoxicity for melanoma cells, and provide a rational basis for selective DDS, as well as targeted cytotoxicity.

As a promising melanogenesis-targeted antimelanoma molecule, we demonstrated that NPrCAP can be a useful tool for developing DDS, chemotherapy, and immunotherapy. Based on that multifunctionality, we further combined NPrCAP with magnetic hyperthermia to develop CTI therapy. From the promising results, preliminary clinical trials for human melanoma patients have started. While larger clinical studies must be conducted to confirm the clinical outcomes, strategies to boost the immune response from CTI therapy are needed to improve patient responses. Melanoma immunotherapies using immune checkpoint blocking antibodies for the PD-1/PD-L1 axis and CTLA-4 axis have become a robust strategy for improving clinical outcomes [121]. Preclinical studies examined combinatorial immunotherapy with nanoparticle-mediated hyperthermia [122]. Recently, Chao et al. combined magnetic hyperthermia with anti-CTLA4 antibody [123], and showed that the administration of anti-CTLA4 antibody after thermal ablation by magnetic hyperthermia induced systemic immunity to inhibit metastasis. These results suggest that the combination of a T cell checkpoint blockade with CTI therapy is a promising potential approach. In addition, the microenvironment of poorly immunogenic tumors contains suppressive myeloid stroma. Toll-like receptors (TLR) trigger innate immune activation, and TLR ligands have the potential to reactivate the tumor microenvironment [124]. Combining a TLR agonist with CTI therapy might induce effector T cells and increase tumor-infiltrating lymphocytes. Novel immunological strategies such as these are needed to propel the regression of poorly immunogenic advanced melanoma.

## Figures and Tables

**Figure 1 cancers-14-05588-f001:**
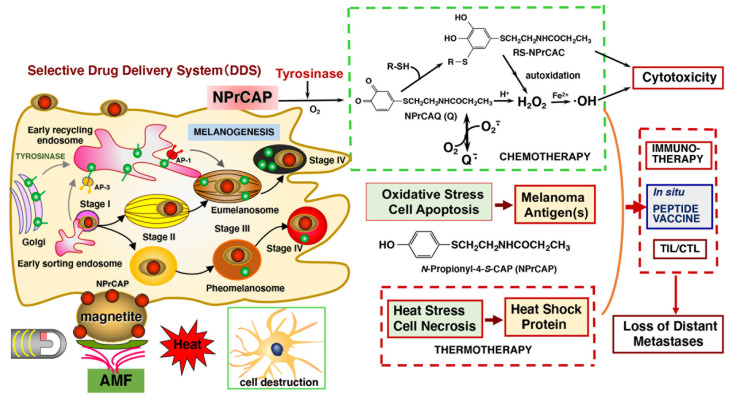
Strategies for chemo-thermo-immunotherapy (CTI therapy) targeting melanogenesis by conjugates of NPrCAP and magnetite nanoparticles upon AMF exposure.

**Figure 2 cancers-14-05588-f002:**
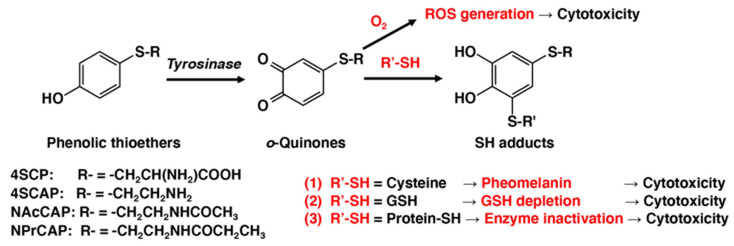
Mechanisms of cytotoxicity of *o*-quinones produced by tyrosinase-catalyzed oxidation of phenolic thioethers.

**Figure 3 cancers-14-05588-f003:**
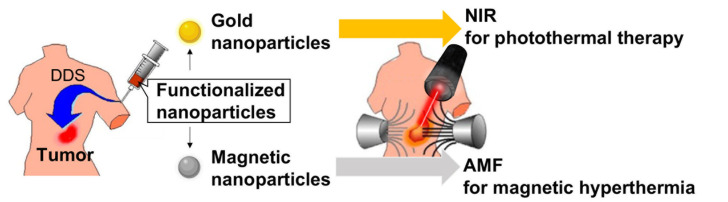
Hyperthermia using nanoparticles. DDS: drug-delivery system, NIR: near-infrared laser, AMF: alternating magnetic field.

**Figure 4 cancers-14-05588-f004:**
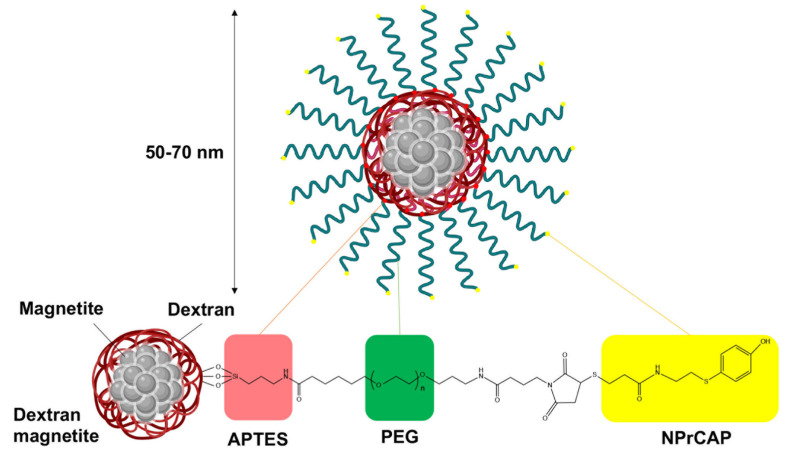
Illustration of NPrCAP/PEG/M. APTES: 3-aminopropyltriethoxysilane, PEG: polyethylene glycol, NPrCAP: *N*-propionyl-4-*S*-cysteaminylphenol.

**Figure 5 cancers-14-05588-f005:**
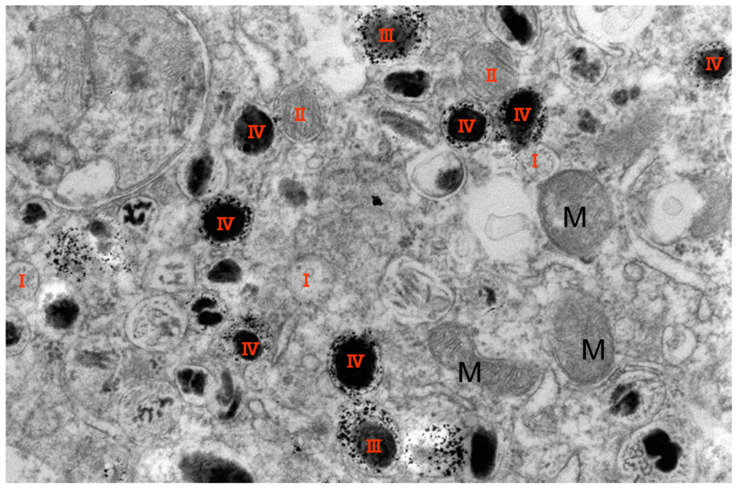
Transmission electron microscope observation. Electron microscopic observation of magnetite nanoparticles in the NPrCAP/M-treated melanoma cells after ip administration to a melanoma-bearing mouse. NPrCAP/M nanoparticles were highly accumulated in the late-stage melanosomes. M: mitochondrion, I-IV: stage of melanosome. Adapted from [45].

**Figure 6 cancers-14-05588-f006:**
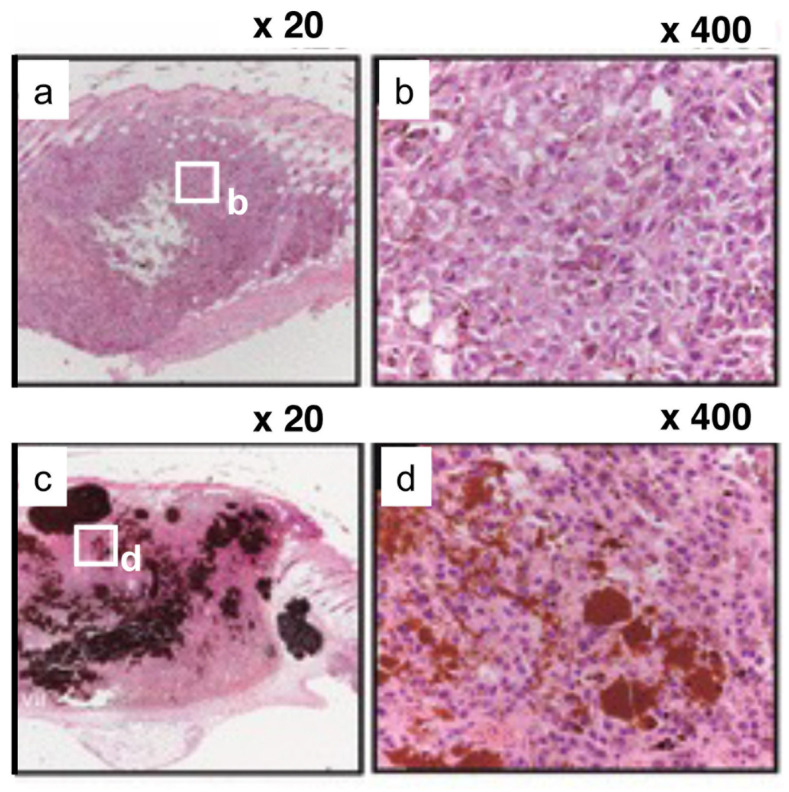
Antitumor effect by magnetic hyperthermia using NPrCAP/M. Pre-established subcutaneous B16-OVA tumors without treatment (**a**,**b**) or magnetic hyperthermia using NPrCAP/M (**c**,**d**) were harvested and analyzed histologically using H&E-stained sections [101]. On Berlin blue staining, brown–black pigments were found to be magnetite particles [27].

**Figure 7 cancers-14-05588-f007:**
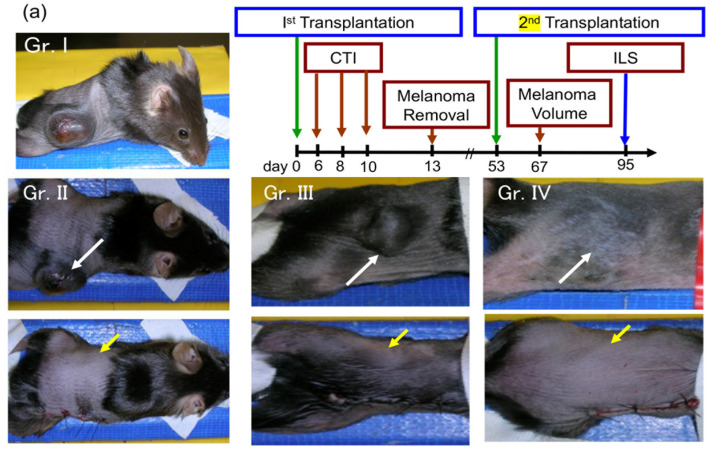
(**a**) Experimental operation processes with mice showing the growth inhibition of re-challenged melanoma (second transplantation) after removal of NPrCAP/M-treated melanoma (first transplantation) through immune processes. Control, non-treated; Gr. I, magnetite alone (no NPrCAP, no heat); Gr. II: Magnetite alone without NPrCAP, but with 43 °C heat for 30 min by AMF exposure; Gr. III: NPrCAP/M treatment, but no AMF exposure; Gr. IV: mouse with both NPrCAP/M and AMF exposure. After removal of the first transplant, all mice received the second re-challenge melanoma transplantation. White arrows, the site transplanted with melanoma at first transplantation; yellow arrows, the site transplanted with rechallenged melanoma at second transplantation. Tumor volume was measured at day 67 and representative photos are shown. ILS: Increased lifespan. CTI: Chemo-thermo-immuno-therapy. Adapted from [45]. (**b**) Treatment protocols for the 4 groups of mice described in (**a**) and an untreated control group. After primary B16F1 melanoma transplantations on day 0, treatments with magnetite nanoparticles with/without NPrCAP and with/without AMF exposure were administered on days 6, 8, and 10, tumors were removed on day 13, and second, re-challenge B16F1 melanomas were transplanted on day 53. (**c**) Daily increases in primary tumor volumes (mm^3^) after transplantation in control, Gr III, and Gr IV mice. (**d**) Tumor volumes (mm^3^) of re-challenge melanomas two weeks after transplantation on day 53, after primary B16F1 transplantations, treatments, and removal of primary tumors, as described in (**b**). (**e**) Kaplan–Meier survival curves of experimental mice up to 90 days after tumor re-challenge. Adapted from [45].

**Figure 8 cancers-14-05588-f008:**
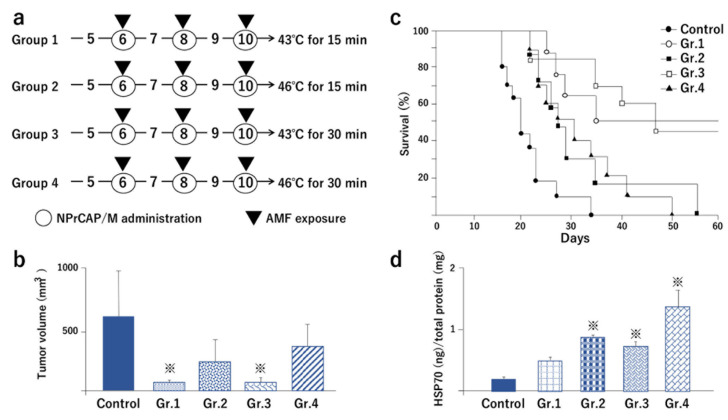
Schedule and conditions of magnetic hyperthermia using NPrCAP/M in treatment of primary B16F1 melanoma-bearing mice and results for second, re-challenge tumor volumes and survival of mice, as well as HSP expression in treated primary tumors. (**a**) Schedule and conditions of NPrCAP/M administration and AMF exposure for Groups 1, 2, 3, and 4 of mice with primary melanoma transplants. Control is the same as described in Figure 7a. (**b**) Tumor volumes on day 14 after re-challenge with B16F1 cells according to the protocol for primary tumor removal and re-challenge described in Figure 7b. (**c**) Kaplan–Meier survival curves up to 60 days after re-challenge. (**d**) Amounts of HSP70 in primary tumors 24 h after the first treatment of magnetic hyperthermia. ^※^ *p* < 0.05 versus control. Modified from [102].

**Figure 9 cancers-14-05588-f009:**
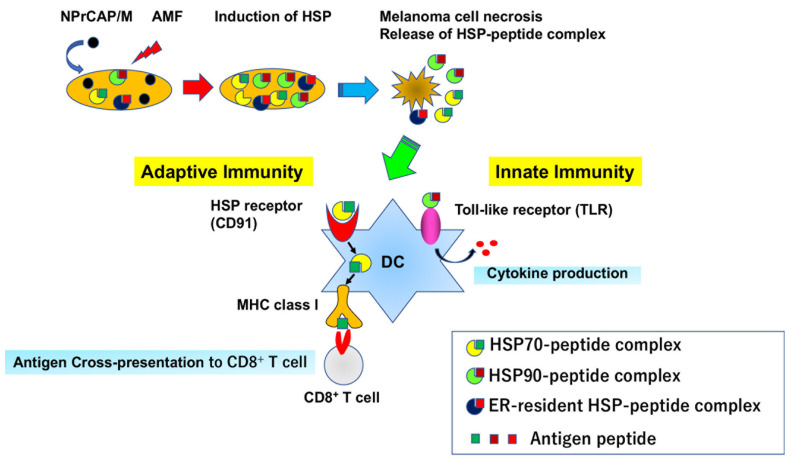
CTI therapy orchestrates the innate and adaptive immunity via production and release of HSP-peptide complex in melanoma cells.

**Table 2 cancers-14-05588-t002:** Effect of dopamine on tumors in patients with malignant melanoma. Modified from [56].

Patient no.	Duration of Treatment (hrs) #	Plasma Level (×10^−5^ M)	Tumor (Percent Labeling Index)	
			Preinfusion	Postinfusion
1	120	4.0	2.0	0.2
2	72	5.2	3.0	0.2
3	48	3.5	1.0	0.1
4	48	3.1	3.0	0.2

**#**: treated at the maximally tolerated dose of 2 μg/kg body weight per minute.

**Table 3 cancers-14-05588-t003:** Chemical structure and property of synthetic Boron compounds related to Dopa. Dopeba: 3,4-dihydroxyphenethylboric acid. Modified from [84].

Chemical Structure (Customary Name)	Abbreviation	MW	^10^B Percent	
			Natural Abundance	92% ^10^B Abundance
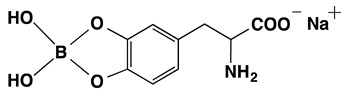	**Dopa borate**	263.00	0.729	3.512
(Sodium Dopa borate)				
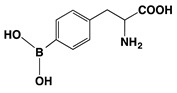	BPA	209.01	0.917	4.423
(*p*-Borono-phenylalanine)				
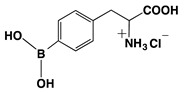	BPA HCl salt	245.48	0.781	3.764
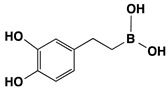	Dopeba	181.99	1.054	5.082

**Table 4 cancers-14-05588-t004:** Clinical results of BNCTT for cutaneous melanoma using BPA.

Reference & No. (Years of Study)	Number of Patients	Melanoma Stage	BPA Dose (mg/kg)	Administration Methods	% Tumor Response (Case Responded)
Fukuda [91] (1987–2002)	22	II–IV	170–210	iv	CR 68.2% (15/22) PR 23.0% (5/22)
Busse [92] (1994–1996)	4	III–IV	400	oral	CR 25% (1/4) PR 50% (2/4)
Menéndez [93] (2003–2007)	7	IV	300	iv	CR + PR 69.3% (overall Survival: 4 to 23 months)
Hiratsuka [94] (2003–2014)	8	II	500	iv	CR 75% (6/8) PR 25% (2/8)

iv: intravenous administration; oral: oral administration. BNCTT (Boron Neutron Capture Thermal Therapy), BPA (Boron phenylalanine), CR (Complete response), PR (Partial response). Modified from [90]. By courtesy of Dr. Fukuda.

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
