# Peer review of "Molecular Events in the Melanogenesis Cascade as Novel Melanoma-Targeted Small Molecules: Principle and Development"

_cancers, 2022, doi:10.3390/cancers14225588_

Round 1
Reviewer 1 Report
In this manuscript, the author described the synthesis of small molecule derivatives based on melanogenesis, reviewed the progress of chemo-thermo-immunotherapy in melanoma.
There are several points need to be improved.
1. In line 118, the author developed three nanoparticles, what are their differences? Why are most animal experiments were carried out by utilizing NPrCAP/M?
2. In Part 4, the author mentioned targeted therapy and immune checkpoint therapy. Is there any evidence that it has synergistic therapeutic effect with selective chemo-thermo-immunotherapy? Or is there evidence that CTI is more effective than targeted therapy and immune checkpoint therapy?
3. As a review, there are only a few recent references.
Author Response
RESPOSE TO REVIEWER 1: 
We wish to express our appreciation to the reviewer for the insightful comments, which have helped us significantly improve the paper.
(x) English language and style are fine/minor spell check required
Response: The paper has been edited and rewritten by an experienced scientific editor, who has improved the grammar and stylistic expression of the paper. In accordance with the reviewer’s comment, the revised paper has been edited again and the spelling has been checked by the experienced scientific editor.
Comment 1: In line 118, the author developed three nanoparticles, what are their differences? Why are most animal experiments were carried out by utilizing NPrCAP/M?
Response: To clarify the differences, we have added the following text in line 128:
We first prepared NPrCAP/MCL to test the combined effects of chemotherapy using NPrCAP and magnetic hyperthermia on melanoma. An in vitro experiment showed that NPrCAP in NPrCAP/MCL had a dose-dependent effect on B16 cell proliferation, and the combination treatment of NPrCAP with magnetic hyperthermia was determined to have an additive effect. Next, we prepared NPrCAP/M by direct conjugation of magnetite nanoparticles with NPrCAP for melanoma-specific DDS. Most of our animal experiments were carried out by utilizing NPrCAP/M. We then synthesized NPrCAP/PEG/M with long chains expecting increased cell binding for clinical human studies [46,48]. The detailed differences between NPrCAP/M and NPrCAP/PEG/M are described in Section 3.2(b).
Comment 2: In Part 4, the author mentioned targeted therapy and immune checkpoint therapy. Is there any evidence that it has synergistic therapeutic effect with selective chemo-thermo-immunotherapy? Or is there evidence that CTI is more effective than targeted therapy and immune checkpoint therapy?
Response: We have not investigated whether CTI therapy is enhanced by targeted therapy and immune checkpoint therapy. Preclinical studies examined combinatorial immunotherapy with nanoparticle-mediated hyperthermia [122]. Recently, Chao et al. combined magnetic hyperthermia with anti-CTLA4 antibody [123], and showed that administration of anti-CTLA4 antibody after thermal ablation by magnetic hyperthermia induced systemic immunity to inhibit metastasis. These results suggest combination of T cell checkpoint blockade with CTI therapy is a promising potential approach. Accordingly, we have added this text in line 853 and two recent references.
[122] Moy, A.J.; Tunnell, J.W. Combinatorial immunotherapy and nanoparticle mediated hyperthermia. Adv. Drug Deliv. Rev. 2017, 114, 175-183. https://doi.org/10.1016/j.addr.2017.06.008
[123] Chao, Y.; Chen, G.; Liang, C.; Xu, J.; Dong, Z.; Han, X; Wang, C.; Liu, Z. Iron nanoparticles for low-power local magnetic hyperthermia in combination with immune checkpoint blockade for systemic antitumor therapy. Nano Lett. 2019, 19, 4287-4296. https://doi.org/10.1021/acs.nanolett.9b00579
Comment 3: As a review, there are only a few recent references.
Response: We added the following sentence containing recent references in line 299.
A number of attempts to increase the efficacy of NAcCAP have been reported. Robins et al. [64-68] synthesized various NAcCAP analogues with the intention of increasing the lipophilicity of the compounds. A modest increase in antimelanoma activity against several melanoma cell lines was observed, which was correlated with increased lipophilicity. However, those compounds also exhibited tyrosinase-independent cytotoxicity against an amelanotic SK-Mel-24 melanoma and an ovarian cell line. Of particular interest is a recent report showing that a hybrid of 4SCAP with triazene, a DNA alkylating compound [69]. Those hybrids were found to be excellent tyrosinase substrates. Some of those compounds were unexpectedly devoid of hepatotoxicity while maintaining cytotoxic activity in melanoma cells. 4SCAP appears to be an important component for the new strategy of developing anti-melanoma agents.
[64] Lant, N.J.; McKeown, P.; Kelland, L.R.; Rogers, P.M.; Robins, D.J. Synthesis and antimelanoma activity of analogues of N-acetyl-4-S-cysteaminylphenol. Anticancer Drug Des. 2000, 15, 295-302. PMID: 11200505
[65] Lant, N.J.; McKeown, P.; Timoney, M.C.; Kelland, L.R.; Rogers, P.M.; Robins, D.J. Synthesis and anti-melanoma activity of analogues of N-acetyl-4-S-cysteaminylphenol substituted with two methyl groups alpha to the nitrogen. Anticancer Drug Des. 2001, 6, 49-55. PMID: 11762644
[66] Pearson, V.C.; Ferguson, J.; Rogers, P.M.; Kelland, L.R.; Robins, D.J. Synthesis and antimelanoma activity of tertiary amide analogues of N-acetyl-4-S-cysteaminylphenol. Oncol Res. 2003, 13, 503-512. https://doi.org/ 10.3727/000000003108748027
[67] Ferguson, J.; Rogers, P.M.; Kelland, L.R.; Robins, D.J. Synthesis and antimelanoma activity of sterically congested tertiary amide analogues of N-acetyl-4-S-cysteaminylphenol. Oncol Res. 2005, 15, 87-94. PMID: 16119006
[68] Nicoll, K.; Robertson, J.; Lant, N.; Kelland, L.R.; Rogers, P.M.; Robins, D.J. Synthesis and antimelanoma activity of reversed amide analogues of N-acetyl-4-S-cysteaminylphenol. Oncol. Res. 2006, 16, 97-106. https//doi.org/ 10.3727/000000006783981206
[69] Granada, M.; Mendes, E.; Perry M.J.; Penetra, M.J.; Gaspar, M.M.; Pinho, J.O.; Serra, S.; António, C.T.; Francisco, A.P. Sulfur Analogues of Tyrosine in the Development of Triazene Hybrid Compounds: A New Strategy against Melanoma. ACS Med. Chem. Lett. 2021, 12, 1669-1677. https://doi.org/10.1021/acsmedchemlett.1c00252.
Accordingly, we have added some recent references (refs. [63][64][65][66][67][68][69][122][123]).
We wish to thank the reviewer again for the valuable comments.
Reviewer 2 Report
The review entitled: "Molecular events in the melanogenesis cascade as novel melanoma-targeted small molecules: principle and development" from Wakamatsu et al is very interesting and I recommend for publication in Cancers with minor revisions.
Please, confirm the veracity of this sentence “Melanin consists of small molecule derivatives that are always synthesized by melanoma cells.” Because there are amelanotic melanoma cells…
L-dopa + L-dopa decarboxylase inhibitor is a combination widely used for Parkinson disease. In this manuscript it is proposed for the treatment of melanoma. Considering what we already know about this drug combination, there are some adverse effects that can be expected (e.g. central nervous system effects) with its use?
The authors mentioned some limitations for 4SCAP, such as its low solubility and adverse effects. Can nanotechnology solve this type of limitation? Some studies have explored the nanoformulation of this compound.
Why Figure 3 is not a table?
In figure 8, which is the mean tumor volume for control group? In the image, the tumor of Gr 1 looks very big. Ethical issues should be always addressed during animal experiments.
Author Response
RESPONSE TO REVIEWER 2:
We wish to express our strong appreciation to the reviewer for the insightful comments on our paper. We feel the comments have helped us significantly improve the paper.
(x) English language and style are fine/minor spell check required
Response: The paper has been edited and rewritten by an experienced scientific editor, who has improved the grammar and stylistic expression of the paper. In accordance with the reviewer’s comment, the revised paper has been edited again and the spelling has been checked by the experienced scientific editor.
Comment 1: Please, confirm the veracity of this sentence “Melanin consists of small molecule derivatives that are always synthesized by melanoma cells.” Because there are amelanotic melanoma cells…
Response: Melanin biosynthesis is present even in amelanotic melanoma. “Amelanosis” reflects simply the macroscopic visibility of color change. Under microscopy brown particulates visualized in amelanotic melanoma cells are, in fact, melanin pigments that can be identified by silver staining. We added the following sentences in abstract.
“Amelanosis” reflects simply the macroscopic visibility of color change (hypomelanosis). Under microscopy melanin pigments and their precursors are present in amelanotic melanoma cells.
Comment 2: L-dopa + L-dopa decarboxylase inhibitor is a combination widely used for Parkinson disease. In this manuscript it is proposed for the treatment of melanoma. Considering what we already know about this drug combination, there are some adverse effects that can be expected (e.g. central nervous system effects) with its use?
Response: The reviewer’s comments are correct. Accordingly following sentences are added in line 198 in the text.
In this animal experiment, a hypercatecholamine-type state was observed [54]. The animals became agitated and tremulous and usually died with 1-2h after administration. These acute toxic effects of L-dopa are probably mediated by its conversion to dopamine by the enzyme, dopadecarboxylase.
Comment 3: The authors mentioned some limitations for 4SCAP, such as its low solubility and adverse effects. Can nanotechnology solve this type of limitation? Some studies have explored the nanoformulation of this compound.
Response: We added the following sentence in line 257.
Recent progress in nanotechnology may overcome the limitations, such as low solubility and adverse effects of compounds [63] and exploring the nanoformulation of 4SCAP is a possible approach. Alternatively, we took the approach of modifying 4SCAP by chemical synthesis.
[63] Liu, W-Y.; Lin, C-C.; Hsieh, Y-S.; Wu, Y-T. Nanoformulation development to improve the biopharmaceutical properties of fisetin using design of experiment approach. Molecules 2021, 26, 3031. https://doi.org/10.3390/molecules26103031
Comment 4: Why Figure 3 is not a table?
Response: We changed Figure 3 to Table 3. Therefore, the figures and table numbers after this Table 3 have changed.
We made minor modifications in Table 4.
Comment 5: In figure 8, which is the mean tumor volume for control group? In the image, the tumor of Gr 1 looks very big. Ethical issues should be always addressed during animal experiments.
Response: In Figure 7, the tumor volume is shown by the mean of tumor volume. There was not any significant difference between the tumor volumes of control, non-treated group and Gr. I group (magnetite alone; no NPrCAP, no heat).
Ethical issues of animal experiments are always obtained. This was not addressed as the results are shown by adaptation of the previous report [45].
We made minor changes to Figures 7b-e and 8b, d. Along with that, we added some explanations of the phrases to the Figure legend.
We wish to thank the reviewer again for the valuable comments.